# Effect of *Saccharomyces boulardii* Supplementation on Performance and Physiological Traits of Holstein Calves under Heat Stress Conditions

**DOI:** 10.3390/ani9080510

**Published:** 2019-07-31

**Authors:** Jae-Sung Lee, Nouali Kacem, Won-Seob Kim, Dong Qiao Peng, Young-Jun Kim, Youn-Geun Joung, Chanhee Lee, Hong-Gu Lee

**Affiliations:** 1Department of Animal Science and Technology, Sanghuh College of Life Sciences, Konkuk University, Seoul 05029, Korea; 2Team of an Educational Program for Specialists in Global Animal Science, Brain Korea 21 Plus Project, Sanghuh College of Life Sciences, Konkuk University, Seoul 05029, Korea; 3Department of Food and Biotechnology, College of Science and Technology, Korea University, Sejong 30019, Korea; 4Eaglevet Co. Ltd., Chungcheongnam-do 32417, Korea; 5Department of Animal Sciences, OARDC, The Ohio State University, Wooster, OH 44691, USA

**Keywords:** *Saccharomyces boulardii*, growth, diarrhea, heat stress, dairy calf

## Abstract

**Simple Summary:**

In this study, the effects of *Saccharomyces boulardii* (SB) supplement on the performance and physiological traits of Holstein calves under heat stress were investigated using a climatic chamber. We revealed that supplementation with SB incorporated into milk replacer can ameliorate the negative impact of heat stress on Holstein dairy calves by increasing dry matter intake (DMI), reducing rectal temperature and heart rate, and alleviating diarrhea via modulating pathogenic bacteria in the digestive tract. The results showed that SB can be used as an alternative anti-stressor in the diet of young dairy calves under heat stress (HS).

**Abstract:**

The objective of this study was to determine the effects of *Saccharomyces boulardii* CNCM I-1079 (SB) as a feed additive on performance, diarrhea frequency, rectal temperature, heart rate, water consumption, cortisol level, and fecal bacteria population in Holstein calves (28 ± 1.6 days of age, body weight of 45.6 ± 1.44 kg, *n* = 16) under thermal neutral (TN) and heat stress (HS) conditions. During the TN period for 21 days (d 1 to 21), calves receiving SB showed quadratic or linear effects compared to the control group, showing higher dry matter intake (DMI, *p* = 0.002), and water consumption (*p* = 0.007) but lower frequency of fecal diarrhea (*p* = 0.008), rectal temperature (*p* < 0.001), heart rate (*p* < 0.001), and fecal microbiota at 21 day (*Escherichia coli*, *p* = 0.025; *Enterobacteriaceae*, *p* = 0.041). Meanwhile, calves exposed to HS for 7 days (d 22 to 28) receiving SB showed quadratic or linear effects compared to the control group, showing higher DMI (*p* = 0.002) but lower water consumption (*p* = 0.023), rectal temperature (*p* = 0.026), and cortisol level (*p* = 0.014). Our results suggest that live SB is useful in the livestock industry as an alternative to conventional medication (especially in times of suspected health problems) that can be added to milk replacer for young dairy calves experiencing HS.

## 1. Introduction

Young calves with diarrhea symptoms show morbidity and poor growth rates. They can increase vaccination costs and mortality, which in turn could lead to severe economic losses for the livestock industry worldwide [1]. A study by Brickell et al. [2] has reported that approximately half of all deaths among calves up to the first month after birth are due to diarrhea. Calf diarrhea has a complex and multifactorial etiology. For any infectious disease, nutritional, environmental, and herd management factors such as housing type, colostrum intake, and hygienic conditions can be associated with field outbreaks, indicating that these factors may influence the severity and outcome of the disease [3,4].

Stressful environmental factors such, as warmer temperature and high humidity can cause heat stress (HS) in calves. Exposure of calves to HS causes poor growth performance due to reduced feed intake [5,6] and impaired homeostatic mechanisms, along with altered physiological status, including endocrine and immune systems [5,7,8]. Therefore, reduced feed intake coupled with lower physiological responses can cause poor growth, outbreak of calf diarrhea, and death in extreme cases.

Addition of 0.5 g of a product containing *Saccharomyces cerevisiae* CNCM I-1077 (2 × 10^10^ CFU/g) to grain can increase dry matter intake (DMI) and weight gain prior to weaning [9]. However, when 1.0 g of a product containing *Saccharomyces boulardii* CNCM I-1079 (SB; 2 × 10^10^ CFU/g) is incorporated into milk replacer, it does not improve DMI or growth of young calves [10]. Interestingly, less calves have diarrhea or pneumonia if they receive yeasts, regardless of dosages or strains [9,10], without noting conditions of ambient temperature and humidity. Calves experiencing HS show elevated body temperature [11,12,13] which can increase intestinal permeability of enteric pathogens. Calves exposed to lipopolysaccharide (LPS; *E*. *coli* O55:B5) also show increased body temperature [14]. Meanwhile, yeast can alleviate the effects of HS on the number of undesirable enteric microbes [14] and acute inflammatory responses in swine [15]. However, few data are available on calf responses to SB coupled with HS in a controlled environment. Therefore, the objective of this study was to determine the effect of SB supplementation in milk replacer on performance and physiological traits of dairy young calves under HS. We hypothesized that serial concentrations of SB ranging from 0.5 to 2.0 g (2 × 10^10^ CFU/g) could reduce responses of calves under HS, such as increasing rectal temperature, heart rate, and frequency of fecal diarrhea.

## 2. Materials and Methods

### 2.1. Animals and Diets

All procedures involving animals were approved by the Institutional Animal Care and Use Committee (IACUC) of Konkuk University (Approval No.: KU16095).

Sixteen Holstein-Friesian male calves (28 ± 1.6 days of age, body weight of 45.6 ± 1.44 kg) were grouped (total 4 groups, 4 calves per group). Calves in each group were randomly assigned to one of four treatments: (1) control, a control milk replacer (Easy Bio System, Inc., Seoul, Korea); (2) low SB (LSB), control milk replacer supplemented with 0.5 g of SB; (3) medium SB (MSB), control milk replacer supplemented with 1.0 g of SB; and (4) high SB (HSB), control milk replacer supplemented with 2.0 g of SB. Because the chamber could house 4 animals at a time, groups were staggered to start the experiment. A dried yeast of SB, a monogastric or young ruminant specific yeast containing 2.0 × 10^10^ CFU/g of *Saccharomyces boulardii* CNCM I-1079 with beyond 99% of alive (Eagle Vet Co., Ltd., Chungcheongnam-do, South Korea), was used in this study. The trial length was 28 d with two environmental periods consisting of 21 d of thermal neutral (TN) conditions and 7 d of HS conditions. Calves were housed on straw bedding in one pen with 4 compartments and 12 feeding buckets in a climatic chamber located at Konkuk University Farm Facility. 

All calves were individually fed a milk replacer twice daily (06:00 h in the morning and 18:00 h in the evening, 2.0 L each time per day). Commercial calf starter, rice straw, and water were offered individually in feeding buckets and recorded before the morning feeding. During both experimental periods, calves were fed *ad libitum*, and residues of each calf starter, rice straw, and water were recorded daily. The calf starter was provided by National Agricultural Cooperative Federation (Konkuk calf starter; Anyang, Korea) to meet nutrition requirements of the National Research Council (NRC) [16] for growing calves. Samples of milk replacer, calf starter, and rice straw were ground to pass a 1-mm screen (standard model 4, Arthur H. Thomas Co., Philadelphia, PA, USA) and chemically analyzed using the standard methods of the association of official analytical chemists (AOAC) [17] for dry matter (135 °C in drying in the oven for 2 h; method 967.03), crude ash (550 °C in an ashing furnace for 6 h; method 942.05), crude protein (Kjeldahl procedure; method 976.06), ether extract (method 920.29), and acid detergent fiber (ADFom, method 973.18). Neutral detergent fiber (aNDFom) was determined, according to Van Soest et al. [18], with the addition of sodium sulfite and heat-stable α-amylase (Type XI-A from Bacillus subtilis; Sigma-Aldrich Corp., St. Louis, MO, USA) and the results were calculated without residual ash. The concentrations of calcium and phosphorus were sequentially determined using an inductively coupled plasma optical emission spectrometer (GBC Integra XL, Australia) (Table 1). Body weights were obtained for all animals immediately before feeding at days 0, 21, and 28.

### 2.2. Climatic Chamber

A climatic chamber capable of housing 4 calves (2.5 m length × 2.5 m width × 3.0 m height/calf) separately was used. Therefore, feed and water were offered individually. The chamber had temperature (18 to 34 °C) and humidity (60 to 100%) controllers to maintain ideal temperature and humidity from 0700 to 1900 h. Ambient temperature (STS31, Sensirion AG, Stäfa, Switzerland) and relative humidity (SHT7, Sensirion AG) sensors were located inside the climatic chambers at 1.5 m above the ground. They recorded temperature and humidity every 10 min during the experiment period. During period 1, all calves were housed under constant TN conditions (temperature, 24.8 ± 1.15 °C; humidity, 63.2 ± 4.34% (temperature-humidity index, THI = 72.8 ± 1.80) with a 12:12 h light: dark cycle). Starting on the first day of period 2 (d 22) until the last day of the current study (d 28), animals were subjected to HS conditions (temperature, 30.2 ± 1.29 °C; humidity, 78.3 ± 3.62% (THI = 82.9 ± 2.31) with a 12:12 h light: dark cycle). Values of THI during period 1 never exceeded 73. During period 2, THI values were maintained above 80. THI was calculated based on ambient air temperature (AT, °C) and relative humidity (RH, %) using the following formula, according to a published study [19]: 0.8 × AT + [RH × (AT − 14.4)] + 46.4. 

### 2.3. Physiological Parameters under Thermal Neutral and Heat Stress

Rectal temperatures and heart rates of individual calves were measured daily at 1700 h. Veterinary thermometers (MC-341-E, Omron Healthcare Inc., Kyoto, Japan) were used to measure rectal temperature. A veterinary stethoscope (3M^TM^ Littmann^Ⓡ^ Mater Classic II^TM^ 1392, 3M Co., Ltd., St. Paul, MN, USA) was used to measure heart rates by counting beats per minute. Fecal scores were measured daily at 0700 h using a four-point scale procedure, as described previously by Larson et al. [20]. Scores regarding the fluidity of faeces were as follows: 1 = normal, 2 = soft, 3 = runny, and 4 = watery. A fecal score of 2 or less was considered normal in this experiment. Subsequently, spot samples of fresh faeces (3 g) from each calf was collected at 0700 h on days 0, 21, and 28. They were transferred into 50 ml sterile tubes and kept at −20 °C until fecal microbiota analysis.

To analyze bacterial quantification in fecal samples, genomic DNA was extracted using a NucliSENS easyMAG instrument (BioMerieux, Inc., Marcyl’Etoile, France) according to the manufacturer’s instructions. Briefly, 100 mg of fecal sample was added to 1 ml of lysis buffer and the mixture was incubated at room temperature for 10 min. Lysed sample was then transferred to the well of a plastic vessel containing 50 µL of magnetic silica and subjected to automatic magnetic separation. Genomic DNA extracted was eluted with 100 µL of elution buffer. Subsequently, real-time polymerase chain reaction (PCR) was performed to quantify yeast, *Lactobacillus*, *Prevotella*, *Enterobacteriaceae*, and *E. coli* using group-specific primer sets [21] as shown in Appendix A. Extracted DNA (2 μL) was transferred into 18 μL of PCR mix consisting of SYBR Premix Ex Taq (Takara Bio Inc., Shiga, Japan,), distilled water, forward and reverse primers, and ROX dye (Takara Bio Inc.,) in a 96-microwell real-time PCR plate. Samples in the plate were analyzed using an ABI 7500 Real Time PCR System (Applied Biosystems, Foster City, CA, USA) with the following reaction conditions: 95 °C for 30 s followed by 40 cycles of 95 °C for 5 s and 60 °C for 34 s. For melting curve analysis, the temperature was decreased from 95 °C to 65 °C at a rate of 0.1 °C/s with continuous acquisition of fluorescence signal intensity. Data were analyzed using ABI 7500 v. 2.3 software (Applied Biosystems). Real-time PCR for quantification of *Salmonella enteritidis* (*S. enteritidis*) and *Clostridium perfringens* (*C. perfringens*) was also performed as described above, except that a TaqMan Universal PCR Master Mix (Applied Biosystems) was used as described previously [22]. PCR conditions were: 50 °C for 2 min, 95 °C for 10 min, followed by 40 cycles of 95 °C for 15 s and 60 °C for 60 s. Values of cycle threshold (Ct) were obtained using ABI 7500 software version 2.3 as described previously [22,23]. 

### 2.4. Blood Parameters under Thermal Neutral and Heat Stress

Before feeding milk replacer at 0600 h to calves, two blood samples were collected from the jugular vein of individual calves on days 0, 21, and 28 into K_2_ Ethylenediaminetetraacetic acid (EDTA)-containing vacutainer (Vacutainer, Becton Dickinson, Franklin Lakes, NJ, USA) and serum plus blood collection tubes (Vacutainer, Becton Dickinson). These tubes were immediately placed on ice and then centrifuged at 2000× *g* for 15 min at 6 °C to separate serum. Serum samples were then frozen at −20 °C until analysis.

Samples of whole blood in K_2_ EDTA-containing vacutainers were used for hematological parameters (white blood cell, red blood cell, lymphocyte, monocyte, granulocyte, hemoglobin, hematocrit, mean corpuscular volume, mean corpuscular hemoglobin, mean corpuscular hemoglobin concentration) using VetScan HM2 (Abaxis, Union City, CA, USA). Subsequently, metabolic parameters (glutamic oxaloacetic transaminase, glutamic pyruvic transaminase, non-esterified fatty acid, blood urea nitrogen, creatine, triglyceride, glucose, and cholesterol) and cortisol concentrations in serum samples were measured using an automated biochemical analyzer (HITACHI Automatic Analyzer Model 7180, Hitachi, Tokyo, Japan) and bovine cortisol enzyme-linked immunosorbent assay test kit (Endocrine Technologies Inc., Newark, CA, USA) according to manufacturers’ instructions. 

### 2.5. Statistical Analysis

Daily intake (expressed in kilogram per day of DMI and water consumption), average daily gain (ADG), fecal status, rectal temperature, and heart rate for each dietary treatment were condensed into mean value. These condensed mean values including cortisol and fecal bacteria density during TN and HS periods were shown separately. Animal was the experimental unit. Effects of the level of SB on performance traits, physiological parameters, and fecal microbiota were tested using linear and quadratic contrasts. For comparing treatment means, data condensed from TN or HS period were separately tested by using the generalized linear model (GLM) procedures of statistical analysis system (SAS) [24], following this model: Y = μ + T_i_ + ε_ij_
where, Y is the dependent variable, μ is the overall mean, T_i_ is the effect of dietary treatments and ε_ij_ is the residual error term. The results are presented as the mean and standard error of the mean (SEM). Differences among dietary treatments within a row with different superscripts were analyzed using Tukey multiple ranged test [24] with effects declared significant at *p* < 0.05.

## 3. Results

### 3.1. Growth Performance under Thermal Neutral (TN) and Heat Stress (HS)

We investigated performance parameters of DMI, water consumption, average daily gain (ADG), and feed efficiency of calves receiving SB during the TN period (d 1 to 21) and the HS period (d 22 to 28). As shown in Table 2, calves receiving SB had a quadratic increase in DMI and water consumption during the TN and HS periods. 

Compared to the control group, the DMI of calves in the MSB group was greater (*p* < 0.05) than those in other groups under TN and HS conditions. Water consumption was greater (*p* < 0.05) in the LSB and MSB groups compared to that in the control group during the TN period. No significant difference in water consumption was found among treatment groups during the HS period. There were no significant differences in ADG or feed efficiency among treatment groups either during TN or HS.

### 3.2. Physiological Parameters under Thermal Neutral and Heat Stress

To investigate the physiological conditions of young calves during periods of TN and HS, we measured the occurrence of diarrhea, rectal temperature, heart rate, and cortisol level of calves receiving SB (Table 3). 

Calves receiving SB had quadratic decreases in occurrence of diarrhea and rectal temperature with a linear decrease in heart rate during TN. Additionally, a linear decrease in rectal temperature was observed in calves receiving SB under HS conditions. Calves supplemented with LSB and MSB had significantly (*p* < 0.01) lower fecal scores during the TN period compared to those in the control group. Calves in all SB groups (LSB, MSB, and HSB) showed lower (*p* < 0.05) fecal scores during the HS period compared to those in the control group. Moreover, rectal temperature and heart rate of calves in SB groups were lower (*p* < 0.01) than those in the control group throughout the whole experimental period, except for calves in the LSB group during the HS period. There was no increase in serum cortisol level in calves receiving HSB supplementation during the HS period, although the cortisol levels in calves supplemented with HSB were higher than those in the control during TN period. Hematological and metabolic parameters in blood samples of calves receiving SB supplementation at 0, 21, and 28 d were also measured in this study. However, there was no significant difference in any variable among treatment groups (data not shown).

### 3.3. Changes in Fecal Microbes under Thermal Neutral and Heat Stress

We also observed changes in fecal microbes from calves receiving SB during periods of TN and HS (Table 4). 

In the present study, there was no significant difference in fecal bacteria among treatment groups at day zero. During the TN period, calves receiving SB showed quadratic or linear effects in fecal microbiota at day 21 (*E*. *coli*, *p* = 0.025; *Enterobacteriaceae*, *p* = 0.041). Compared to the control group, fecal densities of *E. coli* in HSB group and *Enterobacteriace* in LSB and HSB groups were lower (*p <* 0.05) during the TN period. However, they were not significantly different among treatment groups during the HS period. As expected, the fecal density of *Saccharomyces spp*. was greater (*p* < 0.05) in calves consuming an SB diet during the both TN and HS periods. During the whole experimental period, *S. enteritidis* or *C. perfringens* was not detected in feces of calves.

## 4. Discussion

HS generally results in decreased feed intake and increased water consumption. It increases body temperature and the incidence of diarrhea by disrupting physiological responses [5,7,8]. In severe cases, it can cause death. Therefore, HS can cause economic loss to the livestock industry. However, several studies have indicated that SB does not improve DMI or BW gain, although SB can result in less frequent occurrence of diarrhea in calves [9,10]. 

We found that DMI of calves in the MSB group was greater (*p* < 0.05) than those in other groups during periods of TN and HS, except for ADG and feed efficiency (Table 2). Although the ADG of calves supplemented with MSB showed a numerical increase compared to others during the experiment period, the increase was not statistically significant. Increased DMI found in the MSB group in the current study was consistent with results in other studies. For example, it has been reported that DMI and body weight (BW) gain in calves are improved when yeast fermentation products are included in calf starter [25]. Water consumption was greater (*p* < 0.05) in the LSB and MSB groups compared to that in the control group during the TN period. No significant difference in water consumption was found among treatment groups during the HS period. Despite the increases in DMI, BW and ADG barely increased in the present study, in contrast to the results of previous studies, after feeding calves with SB incorporated into milk replacer [9] or by oral infusion [10]. These results suggest that nonfermenting yeast can improve DMI without affecting BW gain or ADG in the growth stage of calves. In the current study, with increasing temperature during the HS period, water intake increased, particularly when the temperature was above 30 °C compared to the TN period. Increased water intake during the HS period is closely associated with body temperature control in ruminants [26], because ruminants can control their body temperatures by discharging sweat through skin. For this reason, when animals are exposed to high temperature and high relative humidity, physiological sweat discharge is elevated to control their body temperature. However, no alleviating effect of SB supplementation on water intake of calves during the HS period was observed in the present study. 

Regarding physiological parameters, when calves were subjected to HS conditions, the fecal status of the control calves gradually became soft. Some even showed diarrhea. However, calves in all SB groups (LSB, MSB, and HSB) showed lower (*p* < 0.05) fecal scores during the HS period compared to those in the control group (Table 3). Rectal temperature and heart rates of calves in SB groups were lower (*p* < 0.01) than those in the control group throughout the whole experimental period except for calves in the LSB group during the HS period. Consistent with previous studies [9,10], the current study also showed that milk replacer supplemented with SB could decrease the rectal temperatures and heart rates of calves, particularly during the HS period. Among hormonal parameters, cortisol concentration has been used as an indicator of stress status in animals [27]. High temperature is a factor that can increase stress of calves, resulting in higher blood cortisol concentrations. For instance, continuous exposure to high heat, or temperatures exceeding 37.7 °C, for at least six hours can cause a significant increase in cortisol levels [11]. In the current study, no significant difference in serum cortisol concentration was found among treatment groups on day zero. No increase in serum cortisol level of calves receiving MSB supplementation during the TN period was observed either, although the cortisol level of calves supplemented with MSB was lower (*p* < 0.05) than that in the control group during the HS period. These results suggest that SB diets can alleviate thermal stress experienced by young calves by reducing blood cortisol level. All values in calves were within ranges under normal physiological conditions throughout the whole experimental period.

In the present study, fecal densities of *E. coli* and *Enterobacteriace* were partially decreased (*p <* 0.05) by SB diet during the TN period, but this had no affect among treatment groups during the HS period (Table 4). During the whole experimental period, *S. enteritidis* and *C. perfringens* were not detected in the faeces of calves. Feeding SB to calves increased (*p* < 0.05) the population of fecal yeast during TN and HS periods, suggesting that SB remaining in the digestive tract could decrease the number of undesirable bacteria through competing for sites of colonization of *E. coli* and *Enterobacteriace* during the TN period. In addition, they might provide stability for gut microbiota by decreasing the number of *E. coli* and *Enterobacteriace* during the HS period. Increasing the body temperature of calves that experience HS [11,12,13] can increase intestinal epithelial permeability responses to enteric pathogens. Additionally, previous studies [28,29] have reported that calves exposed to LPS (*E*. *coli* O55:B5) show increased rectal temperature and body temperature compared to saline-infused calves. These changes are accompanied by the presence of fever, the main indicator of acute inflammatory response. Yeasts including SB possess the ability to reduce the number of undesirable enteric microbes [14]. They can inhibit inflammation associated with immune and cortisol responses in swine [15]. The findings of the current study suggesst that the alleviating effect of SB not only reduced *E*. *coli* density in faeces and cortisol level in blood, but also elevated yeast remaining in faeces, leading to decreased rectal temperature and diarrhea frequency in calves experiencing HS. 

All concentrations of SB had linear and quadratic effects on DMI, diarrhea status, rectal temperature, heart rate, cortisol, and fecal bacteria densities of calves during the TN and HS periods. However, the MSB dosage, which could alleviate responses of HS, was apparently more beneficial than the HSB dosage or LSB. We found some supporting data [9,10] on certain concentrations of SB incorporated into milk replacer (0.5 g or 1.0 g of SB; 2 × 10^10^ CFU/g). Indeed, data supporting the determination of the optimal concentration of SB in pre-ruminants, especially under HS conditions using a climatic chamber, are limited.

## 5. Conclusions

Supplementation of SB in milk replacer can lower rectal temperature, heart rate, and potential incidence of diarrhea in Holstein dairy calves under HS. Although feeding SB increased DMI, ADG and feed efficiency were not affected by the SB supplement. Our results suggest that supplementing SB in milk replacer has the potential to improve the health status of calves under TN and HS. Therefore, SB might be useful as an alternative anti-stressor in the diets of young dairy calves under HS.

## Figures and Tables

**Table 1 animals-09-00510-t001:** Chemical compositions of basal diets used in this study.

Basal Diets			
Milk Replacer	Calf Starter		Rice Straw
[% of dry matter basis]			
Crude protein	17.7	23.1	10.3
Ether extract	13.0	9.1	2.9
Crude fiber	7.9	0.0	27.6
Acid detergent fiber (ADFom)	11.0	0.0	46.9
Neutral detergent fiber (aNDFom)	26.6	0.0	78.4
Calcium	1.0	0.6	0.2
Phosphorus	0.6	0.6	0.2

**Table 2 animals-09-00510-t002:** Performance parameters of dry matter intake (DMI), water intake, average daily gain (ADG), and feed efficiency of calves receiving *Saccharomyces boulardii* (SB) during the thermal neutral (TN) period (d 1 to 21) and the heat stress (HS) heat stress (HS) period (d 22 to 28).

	Treatments				SEM	L ^1^	Q
Period	Control	LSB	MSB	HSB			
DMI, kg/d							
TN	1.13 ^b^	1.14 ^b^	1.34 ^a^	1.13 ^b^	0.051	0.793	0.002
milk replacer ^2^	0.5	0.5	0.5	0.5	-	-	-
calf starter	0.59 ^b^	0.61 ^b^	0.81 ^a^	0.60 ^b^	0.043	0.648	0.003
rice straw	0.04	0.04	0.03	0.03	0.002	0.891	0.068
HS	1.68 ^b^	1.55 ^b^	1.91 ^a^	1.64 ^ab^	0.071	0.861	0.002
milk replacer	0.5	0.5	0.5	0.5	-	-	-
calf starter	1.15 ^b^	1.01 ^b^	1.37 ^a^	1.11 ^b^	0.024	0.764	0.003
rice straw	0.03	0.04	0.04	0.03	0.001	0.871	0.064
Water consumption, kg/d							
TN	3.23 ^c^	3.71 ^a^	3.68 ^b^	2.48 ^d^	0.094	0.766	0.007
HS	5.19	4.68	5.66	3.97	0.213	0.572	0.023
ADG, kg/d							
TN	0.45	0.41	0.62	0.38	0.047	0.962	0.057
HS	0.59	0.52	0.63	0.57	0.949	0.807	0.054
Feed efficiency ^3^							
TN	2.51	2.72	2.16	2.97	0.034	0.457	0.060
HS	2.85	2.98	3.03	2.88	0.045	0.562	0.058

Data were expressed as mean during entire TN or HS period followed by generalized linear model (GLM) procedure of statistical analysis system (SAS) (n = 4). ^a,b^ Values within a row with different superscripts differ significantly (*p* < 0.05). ^1^ L: linear, Q: quadratic. ^2^ Recommended allowance: 125 g milk replacer powder in 1 liter of milk. ^3^ Feed efficiency was calculated by dividing DMI (sum of milk replacer, calf starter, and rice straw) by ADG.

**Table 3 animals-09-00510-t003:** Physiological parameters of diarrhea status, rectal temperature, heart rate, and cortisol level of calves receiving SB during the TN period (d 1 to 21) and HS period (d 22 to 28).

	Treatments				SEM	L ^1^	Q
Period	Control	LSB	MSB	HSB			
Occurrence of diarrhea							
TN	1.5 ^a^	1.1 ^b^	1.1 ^b^	1.4 ^ab^	0.03	0.776	0.008
HS	2.0 ^a^	1.2 ^b^	1.2 ^b^	1.1 ^b^	0.06	0.046	0.299
Rectal temperature, °C							
TN	39.0 ^a^	38.5 ^b^	38.8 ^b^	38.9 ^b^	0.02	1.000	<0.001
HS	39.5 ^a^	39.2 ^b^	39.1 ^b^	39.2 ^b^	0.08	0.026	0.306
Heart rate, beats per min							
TN	96.7 ^a^	91.7 ^b^	91.4 ^b^	87.6 ^b^	0.45	<0.001	0.615
HS	108.5 ^a^	106.4 ^ab^	103.3 ^bc^	99.8 ^c^	0.87	0.067	0.219
Cortisol, ng per mL							
d 21	14.5 ^b^	22.7 ^b^	24.2 ^b^	44.6 ^a^	3.70	0.033	0.378
d 28	39.8 ^a^	28.2 ^ab^	28.2 ^b^	30.6 ^ab^	4.30	0.713	0.014

Data were expressed mean during entire TN or HS period followed by GLM procedure of SAS (n = 4). ^a,b^ Values within a row with different superscripts differ significantly (*P* < 0.05). ^1^ L: linear, Q: quadratic.

**Table 4 animals-09-00510-t004:** Changes in fecal microbes of calves receiving SB during the TN period (d 1 to 21) and HS period (d 22 to 28).

	Treatments				SEM	L ^1^	Q
Period	Control	LSB	MSB	HSB			
*E. coli*, Log CFU per 100 mg of faeces (unless otherwise noted)							
d 0	3.9	4.0	3.2	3.7	0.19	0.461	0.055
d 21	3.9 ^a^	2.8 ^ab^	3.5 ^a^	2.4 ^b^	0.22	0.025	0.378
d 28	3.5	2.5	2.6	2.4	0.21	0.069	0.233
*Enterobacteriaceae*							
d 0	5.6	5.4	4.9	5.2	0.22	0.447	0.243
d 21	5.5 ^a^	3.6 ^b^	5.0 ^a^	3.8 ^b^	0.27	0.105	0.041
d 28	4.9	3.5	4.0	3.7	0.23	0.090	0.239
*S. enteritidis*							
d 0	ND ^2^	ND	ND	ND	-	-	-
d 21	ND	ND	ND	ND	-	-	-
d 28	ND	ND	ND	ND	-	-	-
*C. perfringens*							
d 0	ND	ND	ND	ND	-	-	-
d 21	ND	ND	ND	ND	-	-	-
d 28	ND	ND	ND	ND	-	-	-
*Lactobacillus*							
d 0	5.3	5.7	4.6	5.9	0.23	0.872	0.052
d 21	5.3	5.9	4.8	4.7	0.28	0.313	0.092
d 28	4.6	5.3	4.8	4.3	0.27	0.658	0.058
*Prevotella*							
d 0	6.6	5.9	6.3	6.3	0.31	0.857	0.203
d 21	7.5	8.0	7.7	7.5	0.17	0.895	0.102
d 28	7.3	7.4	7.7	7.1	0.28	0.970	0.061
*Saccharomyces* spp.							
d 0	4.0	3.8	4.4	4.3	0.19	0.381	0.059
d 21	4.0 ^b^	5.3 ^a^	4.9 ^a^	5.4 ^a^	0.19	0.013	0.388
d 28	3.8 ^b^	5.3 ^a^	5.4 ^a^	5.0 ^a^	0.22	0.032	0.330

Data were expressed mean during entire TN or HS period. followed by GLM procedure of SAS (*n* = 4). ^a,b^ Values within a row with different superscripts differ significantly (*p* < 0.05). ^1^ L: linear, Q: quadratic. ^2^ ND: not detectable.

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
