# Peer review of "Effect of Saccharomyces boulardii Supplementation on Performance and Physiological Traits of Holstein Calves under Heat Stress Conditions"

_animals, 2019, doi:10.3390/ani9080510_

Round 1
Reviewer 1 Report
This manuscript reported the effect of SB supplement on growth performance under thermal neutral condition and growth and physiological responses under heat stress condition. The aim of this study is important for establishing healthy calf production under heat stress condition.
In this study, calves were kept under thermal neutral condition for 21 days from start of experiment at 28 ±1 days age. Subsequently, the calves were kept under heat stress condition for 7 days from 49 days of age (it is estimated the days of age at start). So, the growth stage of calves was different between thermal neutral and heat stress period. The calves were given milk replacer and solid feed, so that rumen would be more developed at the heat stress period, as DMI during this period was higher than that during thermal neutral period. It means that the effect of growth stage and heat stress is confounded. Author tried to analyze data of thermal neutral and heat stress period separately. Not only the thermal condition but also growth stage was different between the two periods. Authors should consider this point and discuss it in discussion section. In addition, the increase in DMI by SB supplement may facilitate rumen development of young calves, even in heat stress condition.
The discussion section seems to be lengthy due to repeated explanation of the results which were already explained in the result section. Please modify the discussion section by reducing repeated explanation.
Other points
Page 1. Abstract. Please add the information of age and number of calves used in this study. In addition, input the length (7 days) of the heat stress period.
Page 2 and 3. Table 1. Fat content of milk replacer seems to be low. Is it normal range of fat in milk replacer in Korea?
Page 3, line 108. The size of climatic chamber. The size (2.5 m x 2.5 m x 3 m) was presumably for one calf. Note that this size was for one calf. Calves were kept in this chamber whole day during whole experimental period (from 28 days of age to end of the experiment)?
Page 3, line 111. The chamber controlled temperature and humidity from 0700 to 1900. How was the environment controlled from 19:00 to 7:00?
Page 3, line 115 and 117. How authors decided this temperature and humidity. Are there any evidence?
Page 4, line181. Table 2. DMI probably represented total daily dry matter intake of milk replacer and solid feed. It is better to show them separately (milk replacer, calf starter and rice straw).
Page 5, line 194-195. This explanation was not followed the result in Table 2.
Page 7, line 233-234. This explanation was not followed the result in Table 4.
Page 8, line 310. What kind of in vitro experiments. For what in vitro experiment will be used.
Author Response
Dear sir. reviewer 1,
Please find enclosed the revised version of our manuscript entitled “Effect of Saccharomyces boulardii supplementation on performance and physiological traits of Holstein calves under heat stress condition”
We trust that we have made all of the changes necessary and that the manuscript is now ready for publication.
Thank you very much for all valuable comments.
Sincerely,
Hong-Gu LEE, the corresponding author (hglee66@konkuk.ac.kr)

Reviewer 2 Report
This paper investigated the effects of Saccharomyces boulardii (SB) supplement on performance and physiological traits in Holstein calves under heat stress(HS) using a climatic chamber. The experimental results show that SB can be used as an alternative of anti-stressor in diet for young dairy calf under HS. Overall, the article is well organized and its presentation is good. However, some minor issues still need to be improved.
1)Please add the method of multiple comparison in section 2.5.
2)How to determine the range of additive contents in three SB supplement treatments?
3)Is there no transition period between period 1 and period 2 ? The results will be more convincing if author can provide the graph of some physiological parameters during period of heat stress .
4)Based on the experimental data, whether it has some dose-effect relation that SB as an alternative of anti-stressor in diet for young dairy calf under HS?
Author Response
Dear sir. reviewer 2,
Please find enclosed the revised version of our manuscript entitled “Effect of Saccharomyces boulardii supplementation on performance and physiological traits of Holstein calves under heat stress condition”
We trust that we have made all of the changes necessary and that the manuscript is now ready for publication.
Thank you very much for all valuable comments.
Sincerely,
Hong-Gu LEE, the corresponding author (hglee66@konkuk.ac.kr)
